

# HONO chemistry at a suburban site during the EXPLORE-YRD campaign in 2018: HONO formation mechanisms and impacts on O₃ production

Can Ye[1], Keding Lu[1], Xuefei Ma[1], Wanyi Qiu[1], Shule Li[1], Xinping Yang[2], Chaoyang
Xue[3], Tianyu Zhai[1], Yuhan Liu[1], Xuan Li[1], Yang Li[1], Haichao Wang[1], Zhaofeng Tan[1],
Xiaorui Chen[1], Huabin Dong[1], Limin Zeng[1], Min Hu[1], Yuanhang Zhang[1]

[1] State Key Joint Laboratory of Environment Simulation and Pollution Control, College of Environmental Sciences and
Engineering, Peking University, Beijing, 100871, China
[2] State Environmental Protection Key Laboratory of Vehicle Emission Control and Simulation, Chinese Research
Academy of Environmental Sciences, Beijing, 100012, China
[3] Max Planck Institute for Chemistry, Mainz 55128, Germany

*Correspondence to*: Keding Lu (k.lu@pku.edu.cn), Yuanhang Zhang(yhzhang@pku.edu.cn)

**Abstract.** HONO is an important precursor for OH radicals that impact secondary pollutants production. However, there are still large uncertainties about different HONO sources, which hinder accurate predictions of HONO concentration and hence atmospheric oxidation capacity. Here HONO was measured during the EXPLORE-YRD campaign, along with other important parameters, enabling us to comprehensively investigate HONO variation characteristics and evaluate the relative importance of different HONO sources by using a box model. HONO showed significant variations, ranging from several tens of ppt to 4.4 ppb. The average diurnal pattern of HONO/NO$_x$ showed a maximum of 0.17 around noon and resembled that of j(O$^1$D), indicating the existence of photo-induced sources. Modeling simulations with only the default HONO source (OH+NO) largely underestimated HONO concentrations, with the modeled averaged noontime HONO concentration an order of magnitude lower than the observed concentration. The calculated unknown source strength (P$_{unknown}$) of HONO showed a nearly symmetrical diurnal profile with a maximum of 2.5 ppb h$^{-1}$ around noon. The correlation analysis and sensitivity tests showed that photo-induced NO$_2$ conversion on the ground was able to explain P$_{unknown}$. Additional HONO sources incorporated into the box model improved the model's performance in simulating HONO concentrations. The revised box model well reproduced nighttime HONO concentration but still underestimated daytime HONO concentration. Further sensitivity tests indicated the underestimation of daytime HONO was not due to uncertainties of photo-induced NO$_2$ uptake coefficients on the ground or aerosol surfaces or enhancement factor of nitrate photolysis but was more likely to other sources that were not considered in the model. Among the incorporated heterogeneous HONO sources and the gas-phase source, photo-induced NO$_2$ conversion on the ground dominated the modeled HONO production during the daytime, accounting for 73% of the total, followed by NO+OH (10%), NO$_2$ hydrolysis on the ground surface (9%),





photo-induced $NO_2$ conversion on the aerosol surface (3%), nitrate photolysis (3%), and $NO_2$ hydrolysis on the aerosol surface (2%). $NO_2$ hydrolysis on the ground surface was the major source of nighttime HONO, contributing to 65% of total HONO production. HONO photolysis contributed to 43% of $RO_x$ production during the daytime, followed by $O_3$
photolysis (17%), HCHO photolysis (14%), ozonolysis of alkenes (12%), and carbonyl photolysis (10%). The net ozone production rate (12.6 ppb h$^{-1}$) with observed HONO as a model constraint decreased by 45% compared to that (6.7 ppb h$^{-1}$) without HONO as a model constraint, indicating HONO evidently enhanced HONO production and hence aggravated $O_3$ pollution in summer seasons. Our study emphasized the importance of $NO_2$ heterogeneous conversion on the ground surface in HONO production and accurate parameterization of HONO sources in predicting secondary
pollutants production.

## 1 Introduction

OH radicals are the primary oxidant in the atmosphere, accelerating the degradation of gas-phase pollutants and regulating the lifetime of trace gases. OH oxidation will lead to secondary pollutants formation like SOA and $O_3$, which both have adverse impacts on air quality. HONO is one of the most important precursors of OH radicals by
photolysis (R1), especially in polluted areas, contributing to up to 92% of primary OH production (Alicke et al., 2003; Kleffmann et al., 2005; Elshorbany et al., 2009; Hou et al., 2016; Xue et al., 2020; Tan et al., 2017). Previous studies reported that HONO was not only an vital OH precursor in the early morning but also throughout the day (Neftel et al., 1996; Ren et al., 2003; Ren et al., 2006; Michoud et al., 2012). Therefore, HONO is closely linked to atmospheric oxidation capacity, and accurate representation of HONO sources in models is important for air quality predictions and
control strategy development.

$$HONO \xrightarrow{h\nu} OH+NO \qquad\qquad\qquad R1$$

$$OH+NO \xrightarrow{M} HONO \qquad\qquad\qquad R2$$

However, HONO sources are still controversially discussed due to the complexity of their formation mechanisms. Till now, HONO sources, including gas-phase reactions, direct emissions, heterogeneous reactions on varied surfaces, and acid displacement reactions, have been proposed to explain HONO. The gas-phase HONO source mainly comes from the reaction between OH and NO (R2). This source is too small to explain high HONO levels frequently observed at
different sites in the world (Lee et al., 2016; Michoud et al., 2014; Liu et al., 2019b; Kleffmann et al., 2003), indicating there still exist other sources. Large HONO unknown sources were reported at different sites, for instance, 3 ppb h$^{-1}$ in Tai'an (Xue et al., 2022), 2.58 ppb h$^{-1}$ in Beijing (Spataro et al., 2013), 0.7 ppb h$^{-1}$ in Paris (Michoud et al., 2014), 1.77 ppb h$^{-1}$ in Santiago (Elshorbany et al., 2009). The unknown source strength was one magnitude higher than the HONO production rate by the OH+NO reaction. Other gas-phase sources, like the reaction between excited $NO_2$ and water



molecules (Li et al., 2008) and the photolysis of nitro-phenols (Bejan et al., 2006), have also been proposed. However, the two gas-phase sources were verified to be of minor importance to HONO production under atmospherically relevant conditions (Yang et al., 2021a; Sorgel et al., 2011). Direct emissions include emissions from combustion sources and fertilized soil by biological processes. HONO could be directly released from vehicle exhausts and is often characterized by the HONO/$NO_x$ ratio, which is typically less than 2% $NO_x$ emissions (Kurtenbach et al., 2001; Xu et

al., 2015; Kramer et al., 2020). Hence, HONO emissions by vehicles were considered to play a minor role in HONO formation, especially in rural and remote areas. Biomass burning and wildfire also contributed to HONO emissions on a global scale (Nie et al., 2015; Cui et al., 2021). HONO emitted by wildfires was estimated to account for two-thirds of OH production in fresh wildfire plumes, which accelerated the age of plumes and $O_3$ production (Theys et al., 2020). In addition, microbe-produced nitrite from fertilized soil can produce HONO (Su et al., 2011; Oswald et al., 2013;

Weber et al., 2015; Scharko et al., 2015; Wang et al., 2021). Recent field studies revealed HONO and concurrent $O_3$ and $H_2O_2$ enhancements after fertilization events in the North China Plain, implying the large impact of soil-emitted HONO on regional atmospheric oxidation capacity (Xue et al., 2021).

$$NO_2+H_2O \xrightarrow{\text{surfaces}} HONO+HNO_3 \qquad\qquad R3$$

$$NO_2+A^{red} \rightarrow HONO+A^{''} \qquad\qquad R4$$

$$HNO_{3(ads)} \xrightarrow{\text{hv}} [HNO_3]^*_{ads} \qquad\qquad R5$$

$$[HNO_3]^*_{ads} \rightarrow HONO_{ads}+O_{ads} \qquad\qquad R6$$

Heterogeneous processes are believed to be important for HONO production. Heterogeneous $NO_2$ conversion on humid surfaces (R3) has long been identified as an important source of nighttime HONO, with two $NO_2$ molecules

converting to one HONO molecule (Finlayson-Pitts et al., 2003; Stutz et al., 2004). However, R3 could not explain HONO levels during the daytime. Field measurements revealed HONO unknown source was typically positively correlated with j($NO_2$), indicating a photo-induced process (Vogel et al., 2003; Lee et al., 2016). Photo-induced $NO_2$ conversion on humic acid was proposed to explain daytime HONO (R4) (Stemmler et al., 2006; Stemmler et al., 2007). This source was found to be important to sustain high daytime HONO levels (Wong et al., 2013; Zhang et al., 2016;

Xue et al., 2022). In addition, heterogeneous nitrate/$HNO_3$ photolysis on varied surfaces was found to be enhanced compared to gas-phase $HNO_3$ and also contributed to HONO formation (Zhou et al., 2011; Ye et al., 2016; Ye et al., 2017; Bao et al., 2018; Baergen and Donaldson, 2013). However, the importance of heterogeneous nitrate/$HNO_3$ photolysis on HONO production (R5-R6) is highly uncertain, with the enhancement factor (EF) for nitrate photolysis relative to gas-phase nitric acid spanning three orders of magnitude from 1 to 1000 (Ye et al., 2017; Bao et al., 2018;

Laufs and Kleffmann, 2016; Romer et al., 2018; Shi et al., 2021). HONO deposition during the nighttime and subsequent desorption from the soil by strong acid displacement during the daytime was proposed to be a HONO



source (Vandenboer et al., 2015). Although varied sources have been proposed, there are still large uncertainties about many mechanisms and the relative importance of each mechanism in different environments.

$O_3$ pollution is serious in China in the summer and becomes a major air quality issue. The observed ozone-increasing

rates from 2013-2017 were 3.11 ppb year$^{-1}$, 2.29 ppb year$^{-1}$, 0.56 ppb year$^{-1}$, and 1.63 ppb year$^{-1}$ in Beijing–Tianjin–Hebei (BTH), Yangtze River Delta (YRD), Pearl River Delta (PRD), and Sichuan Basin (SCB), respectively, the four megacity clusters in China (Li et al., 2019a). $O_3$ is produced by complex photochemical chain reactions between VOCs and $NO_x$ in the presence of sunlight. Now the control strategy for $O_3$ pollution mainly focuses on the reduction of VOC and $NO_x$, which mainly depend on the $O_3$ formation regimes. HONO photolysis is one of the major sources of primary

$RO_x$ radicals, which initiate the photochemical reactions leading to $O_3$ formation. More and more field studies and modeling studies found that high HONO levels in China aggravated $O_3$ pollution. For instance, the incorporation of proposed HONO sources improved the predictions of $O_3$ by the WRF-Chem model and increased $O_3$ concentrations by 6-12% compared with cases without HONO sources (Zhang et al., 2016). Fu et al. (Fu et al., 2019) found that newly-added HONO sources increased nitrate concentrations by 17 μg m$^{-3}$ and $O_3$ by 24 ppb during a winter pollution

episode in the PRD of China. Furthermore, missing representations of HONO sources in box models or chemical transport models would affect the judgment of $O_3$ formation regimes and lead to ineffective $O_3$ mitigation strategies (Li et al., 2018b; Liu et al., 2023). Therefore, an accurate understanding of HONO formation mechanisms is important for the refinement of the $O_3$ pollution control policy.

YRD is one of the most populated and polluted areas in the world. Recently, this area has witnessed an evident

increase in $O_3$ levels, with $O_3$ pollution days nearly doubling (28 days to 76 days) from 2014 to 2017 (Liu et al., 2020). Large numbers of studies were carried out that mainly focused on the study of $O_3$ pollution characteristics and determination of $O_3$ sensitivity to VOCs and $NO_x$ (Ding et al., 2013; Xing et al., 2017; Wang et al., 2019), while research on HONO formation mechanisms and their effects on $RO_x$ (OH, $HO_2$, $RO_2$ radicals) budget and $O_3$ production was still limited. The EXPLORE-YRD campaign (EXPeriment on the eLucidation of the atmospheric

Oxidation capacity and aerosol foRmation, and their Effects in the Yangtze River Delta) was carried out with the aim of exploring the atmospheric oxidation capacity in the area and developing an effective co-control strategy for both $O_3$ and particulate matter. As mentioned above, HONO chemistry is closely linked to atmospheric oxidation capacity and, therefore, should be comprehensively studied. In this study, HONO was measured along with other gas-phase (including OH, $HO_2$, VOCs, $NO_x$, etc.), aerosol-phase, and meteorological parameters during the EXPLORE-YRD

campaign in China, allowing us a detailed evaluation of HONO chemistry by employing an observation-based box model. We first investigate the HONO variation characteristics and the unknown source strength. Then correlation analysis and sensitivity tests were performed to investigate the relative importance of different HONO sources preliminarily. Then we incorporate proposed HONO sources into the box model to investigate if these sources could





explain observed HONO concentration. The comparison of modeled and measured HONO provides unique information for understanding different HONO formation mechanisms and exploring their respective contributions to HONO production. Based on modeling results, the contribution of HONO photolysis to primary $RO_x$ production was calculated. In addition, the $O_3$ formation rates calculated with and without HONO sources were also compared to investigate the effects of HONO on $O_3$ production.

## 2 Experimental

### 2.1 Site description

EXPLORE-YRD campaign was performed from May 14 to June 20, 2018 at a suburban site (32.56°N, 119.99°E) in the YRD region, one of the most polluted areas in China. The measurement site is located in the park of the meteorological radar station in Taizhou City, which is situated 200 km northwest of Shanghai City (Figure 1). This site was surrounded by farmlands and fishponds and far away from industrial areas, occasionally influenced by biomass-burning events. Winds from the east and southeast prevailed during the measurement periods, indicating this site is influenced by anthropogenic pollution from the YRD region. All the instruments were placed in five containers, and the sampling inlet was about 5 m above the ground.

### 2.2 HONO measurement

HONO was measured by a commercial HONO monitor (LOPAP, QUMA), which was based on a wet chemical technique. A detailed description of the instrument has been introduced in previous studies (Heland et al., 2001; Xue et al., 2020). Briefly, HONO was sampled by stripping solution (sulfanilamid + HCl) in a stripping coil. Then the sampled solution will react with n-(1-naphthyl)-ethylenediamine-dihydrochloride solution, yielding azo dye, which will be detected by an optical absorption spectrometer at 550 nm. In order to minimize the effect of interfering species, two stripping coils in series were employed. In the first coil, nearly all HONO and a small fraction of interfering species was sampled. In the second coil, assuming the same amount of interfering species was sampled, the difference in signals between the two coils represents the true HONO concentration in the atmosphere. Zero calibration was performed every day. The detection limit and uncertainty of the instrument were 5 ppt and 10%, respectively.

### 2.3 Other measurements

A comprehensive set of gas-phase and aerosol-phase parameters has been measured in the campaign, and detailed information on parameters and instruments is listed in Table S1 in the supporting information. Here we will make a brief introduction. $O_3$, CO, $SO_2$, and $PM_{2.5}$ were measured by Model 49i, Model 48i, Model 43i and Model 1400A



from Thermo Inc, respectively. $NO_x$ was measured by a trace-level analyzer (Model 42i), which was equipped with a home-built photolytic converter for true $NO_2$ measurement to avoid interference from other $NO_y$ species like PAN, HONO, and organic nitrate. As $NO_2$ was an important precursor of HONO, true $NO_2$ measured was used throughout the paper to better understand HONO formation mechanisms. VOCs were measured by an online gas chromatograph equipped with a flame ionization detector and mass spectrometry. HCHO was measured by the Hantzsch fluorescence technique. OH and $HO_2$ radicals were measured by PKU-LIF, and detailed information about PKU-LIF equipment can be found in our previous studies (Lu et al., 2013; Tan et al., 2017; Ma et al., 2022). Photolysis frequencies were calculated using spectroradiometer-measured integrated actinic flux. RH, temperature, pressure, wind speed, and wind direction were simultaneously measured by a portable weather station. The aerosol surface area ($S_a$) was derived from data measured by a scanning mobility particle sizer instrument (SMPS, TSI 3936) and an aerosol particle sizer instrument (APS, TSI 3321). Aerosol-phase water-soluble ions were measured by a gas and aerosol collector equipped ion chromatography (GAC) instrument.

**2.4 Observation-based box model**

HONO concentration was simulated by a box model based on the RACM2-LIM1 mechanism. The model was constrained with measurements of NO, $NO_2$, $O_3$, CO, $H_2O$, C2-C12 VOCs, HCHO, OH radicals, temperature, and pressure with a time resolution of 5 min. Measured HONO was only constrained when try to explore the contribution of HONO to primary $RO_x$ budget and $O_3$ production rate. In other cases, HONO was not constrained in model simulations. In addition, $j(O^1D)$, $j(HONO)$, $j(H_2O_2)$, $j(NO_2)$ and $j(HCHO)$ were also constrained to the model. $CH_4$ and $H_2$ were assumed to be 1900 ppb and 550 ppb, respectively. In RACM2, based on the reactivities with OH, different VOCs were assigned to different lumped species rather than treated individually. A first-order dilution loss term with a lifetime of 8 hours was assigned to all species to represent deposition and advection loss. The modeled PAN and observed PAN matched well (Figure S1), and the ratio of modeled to observed PAN concentration was 1.09 if this dilution term was incorporated. Detailed information on the box model can be found in our previous studies focusing on simulating OH and $HO_2$ radicals (Tan et al., 2017; Tan et al., 2018; Ma et al., 2019).

**2.5 Ozone production rate calculation**

The total ozone production rate ($F(O_3)$) was directly determined by the reactions between NO and peroxy radicals (Eq. 1). Due to a lack of measurements of different peroxy radical species, model-calculated peroxy radical concentrations were used in Eq. 1. $O_3$ was mainly consumed by photolysis and reactions with alkenes, OH, and $HO_2$. In addition, the reaction between OH and $NO_2$ forms $HNO_3$, which constitutes a portion of the $O_3$ loss. $NO_3$ radicals, formed by the reaction between $NO_2$ and $O_3$, could subsequently form $N_2O_5$ or organic nitrate formation, contributing to $O_3$ loss. The





loss rate of $O_3$ can be described in Eq. 2. Therefore, the net ozone production rate $P(O_3)$ can be obtained by calculating the difference between $F(O_3)$ and $D(O_3)$ (Eq. 3).

$$F(O_x)=k_{NO+HO_2}[NO][HO_2]+\sum_i k_{RO_{2i}+NO}[RO_2]_i[NO] \tag{1}$$

$$D(O_x)=k_{O^1D+H_2O}[O^1D][H_2O]+[O_3]\left(k_{O_3+Alkenes}[Alkenes]+k_{O_3+HO_2}[HO_2]+k_{O_3+OH}[OH]\right)+3\left(k_{O_3+NO_2}[NO_2][O_3]-\right.$$

$$\left. k_{NO+NO_3}[NO_3][NO]-j_{NO_3}[NO_3]\right) \tag{2}$$

$$P(O_x)=F(O_x)-D(O_x) \tag{3}$$

## 3 Results and discussion

### 3.1 Overview of the measurements

Figure 2. presents the time profiles of HONO and related chemical species as well as meteorological conditions from May 23 to June 18, 2018. The temperature ranged from 15 ℃ to 35 ℃, and the relative humidity ranged from 30% to 100% during the measurement period. Rainfall events occurred on May 25, 26, and June 10, and corresponding concentrations of both primary and secondary pollutants were low as a result of low solar radiation intensity and the scouring effect of rain. The daily maximum values of $j(O^1D)$ frequently exceeded $2\times10^{-5}$ $s^{-1}$, indicating strong solar radiation intensity during most days of the measurement period. The mean diurnal profiles of HONO and related parameters are shown in Figure 3. The maximum diurnal averaged HCHO concentration was 5 ppb, indicating VOCs at the observation site were abundant. In addition, similar to CO, peak HCHO concentration occurred around 8:00 LT, indicating the effect of anthropogenic emission-related sources. While the sampling site is located in a suburban area with no heavy traffic nearby, the maximum values of NO and $NO_2$ during the campaign reached 38.6 ppb and 49.6 ppb, respectively. The maximum diurnal averaged concentrations of NO and $NO_2$ were 4.4 ppb and 14.8 ppb, respectively. As VOC and $NO_x$ are important precursors of $O_3$, relatively high $NO_x$ and VOCs are conducive to $O_3$ production. $O_3$ concentrations throughout the observation period frequently exceeded Class-II limit values (160 μg $m^{-3}$, which is equivalent to 82 ppb at 298 K and 1013 kpa) of the National Ambient Air Quality Standard, and the highest concentration can reach as high as 150 ppb, indicating serious photochemical pollution. From 6:00 LT, $O_3$ starts to rise rapidly with the increase in solar radiation intensity and reaches a maximum value of 85 ppb at 15:00 a.m. Subsequently, $O_3$ concentrations start to decrease rapidly due to the decrease in $O_3$ in-situ production, NO titration, and dry deposition. Concurrent $PM_{2.5}$ and $O_3$ pollution occurred from May 28 to 30, which is not unexpected as both pollutants are closely linked to the oxidation of VOCs and $NO_x$. The co-occurrence of $PM_{2.5}$ and $O_3$ pollution in China has been frequently observed in recent years (Li et al., 2019b), especially in the spring and autumn seasons, making it a significant challenge for the next phase of air-cleaning actions in China to synergistically control both pollutants. OH





radicals showed a typical diurnal pattern, with a maximum around noon and very low concentrations in the morning and night. The mean diurnal pattern shows that the average peak concentration of OH radicals is $1.0 \times 10^7$ cm$^{-3}$. With respect to summer OH concentrations in China, the average peak OH concentrations measured in this study are higher than that measured in Shanghai ($2.7 \times 10^6$ cm$^{-3}$) (Zhang et al., 2022a), Wangdu ($9.0 \times 10^6$ cm$^{-3}$) (Tan et al., 2017),

Heshan ($3.2 \times 10^6$ cm$^{-3}$) (Ma et al., 2022), Yufa ($4.5 \times 10^6$ cm$^{-3}$) (Lu et al., 2013). The relatively high OH concentration implied strong atmospheric oxidizing capacity in this region. The reaction between OH and NO is the dominant gas-phase source of HONO. Compared with most other studies aiming at investigating the HONO budget, the concurrent measurement of OH radicals in our study will enable us to more accurately assess the contribution from other sources (excluding NO+OH) to HONO production.

HONO concentrations exhibited significant day-to-day variations, with the highest concentration reaching 4.4 ppb and the lowest concentration of several tens of ppt. The maximum HONO concentration was observed on the night of May 24, with HONO increasing from 1.8 ppb to 4.4 ppb in three hours. Interestingly, during this period, PM$_{2.5}$ increased concurrently from 112 μg m$^{-3}$ to a maximum of 203 μg m$^{-3}$, indicating HONO and PM$_{2.5}$ might be originated from the same source. Considering HONO, PM$_{2.5}$, and CO increased concurrently in such a short period, a primary source like

biomass burning was possibly the reason for this event.

In Figure 3, HONO showed a typical diurnal pattern, i.e., high concentrations in the early morning and evening and low concentrations during the daytime, as typically observed at other rural, suburban, and urban sites (Li et al., 2010; Michoud et al., 2014; Xu et al., 2015; Lee et al., 2016). HONO generally began to accumulate in the evening, when HONO photolysis loss ceased. High NO$_2$ concentrations, which facilitated NO$_2$ heterogeneous reactions on humid

surfaces, together with the primary vehicle emissions, led to an increase of HONO during the nighttime. After sunrise, HONO showed a rapid decrease due to fast photolysis and reached a minimum in the later afternoon. Considering that the atmospheric lifetime of HONO is only 10-20 min (with respect to photolysis), however, the averaged noon-time HONO concentration was relatively high (0.5 ppb), which implied the existence of strong daytime HONO sources to counteract its rapid photolysis. Furthermore, it can be clearly seen that periods with high NO$_2$ concentrations were

typically accompanied by high HONO concentrations (May 29-30, June 5-7), and vice versa. This phenomenon further confirmed that NO$_2$ is an important precursor for HONO production. In addition, O$_3$ pollution episodes also coincided with high HONO levels (e.g., June 5-8). This phenomenon can be attributed to the following two reasons: firstly, O$_3$ and HONO have a common precursor NO$_x$; secondly, the photolysis of HONO will lead to the production of OH radicals, which will eventually promote the production of O$_3$, so the higher HONO concentration will lead to the

higher concentration of O$_3$ production, which will be further confirmed in our later model analysis.

HONO/NO$_x$ is an important parameter to characterize the HONO concentration and the extent of NO$_2$ heterogeneous conversion to HONO because it is less affected by transport and up-down convection compared to the HONO





concentration. As shown in Figure 3, HONO/NO$_x$ started to increase after 18:30, when HONO photolysis loss ceased. However, with the fresh NO$_x$ continuing to be emitted, the ratio then remains nearly constant overnight. The
atmospheric lifetime of HONO is very short compared to NO$_x$. The main source of nocturnal HONO was generally considered to be the NO$_2$ heterogeneous hydrolysis reaction, and if this source also dominated daytime HONO production, the rapid photolytic loss of HONO would make the daytime HONO/NO$_x$ value decrease to a minimum. On the contrary, a daytime HONO/NO$_x$ maximum (0.17) was observed around 12:00 LT, indicating other important sources, which is much faster than the NO$_2$ heterogeneous hydrolysis reaction, dominating daytime HONO production.
Additionally, the mean diurnal pattern of HONO/NO$_x$ resembled that of j(O$^1$D), pointing to the possibility of light-promoted HONO sources being dominant during the day. In terms of mean diurnal profiles, the average peak value of HONO/NO$_x$ in this study is much larger than that reported in Beijing (0.08) (Gu et al., 2022), London (0.04) (Lee et al., 2016), and Nanjing (0.055) (Liu et al., 2019a) which indicated the high efficiency of NO$_x$ to HONO conversion. HONO/NO$_x$ measured in summer is typically higher than that measured in winter, which could be attributed to higher
solar radiation intensity and hence promote the conversion of NO$_x$ to HONO by photosensitized reduction. In addition, the high HONO/NO$_x$ in our study may be partially explained by direct HONO emissions around the sampling sites.

The average HONO concentration during the YRD-EXPLORE campaign is 0.62±0.49 ppb, which is within the range of the summer HONO concentration reported in previous studies, and is much lower than the observed concentration in winter in other sites in China, like Wangdu (1.8±1.4 ppb) (Xue et al., 2020), Xianghe (2.18±1.95 ppb) (Zhang et al.,
2022c), Jinan (1.35 ppb) (Li et al., 2018a), and Beijing (0.98±0.85 ppb) (Zhang et al., 2022d), probably due to the relatively low concentration of NO$_x$ in summer. However, the observed HONO concentration is higher than that observed in summer in some suburban sites, like Hongkong (0.35±0.30 ppb) (Xu et al., 2015), Xi'an (0.51 ppb) (Huang et al., 2017), which also indicates the relatively high atmospheric oxidizing capacity in this region. Liu et al. (Liu et al., 2019a) reported a comparable summer HONO concentration of 0.56 ppb at a suburban site in the YRD
region. Relatively high average HONO concentration of 0.76 ppb (0.01-5.95 ppb) was reported at a suburban site in the YRD region during biomass burning season, which was attributed to the influence of biomass events (Nie et al., 2015).

## 3.2 HONO unknown source strength

More and more studies now confirm that gas-phase reactions are far from sufficient to explain the observed HONO
concentrations, and in this study we also calculated the contribution of non-gas-phase reactions to the HONO production based on the measured data (P$_{unknown}$). The reaction between NO and OH is the only considered gas-phase HONO source, and the removal pathways of HONO include photolysis, the reaction with OH radicals. Thus, P$_{unknown}$ is calculated as follows:





$$\frac{dHONO}{dt}=P_{NO+OH}+P_{unknown}-P_{HONO+OH}-P_{photolysis}$$

$$P_{unknown}=\frac{\Delta HONO}{\Delta t}+k_{HONO+OH}[HONO][OH]+j_{HONO}[HONO]-k_{NO+OH}[NO][OH]$$

dHONO/dt represents the change rate of observed HONO at 5-minute intervals. OH, NO, and j(HONO) were

constrained by measured data. The reaction rate constants for NO+OH and HONO+OH were $9.8\times10^{-12}$ cm$^3$ molecules$^{-1}$ s$^{-1}$ and $6.0\times10^{-12}$ cm$^3$ molecules$^{-1}$ s$^{-1}$, respectively. The calculated mean diurnal profile of $P_{unknown}$ is shown in Figure

4. $P_{unknown}$ showed a symmetrical diurnal trend and peaked around 10:00-12:00 with a maximum of 2.5 ppb h$^{-1}$. The

variation of $P_{unknown}$ tracked the profile of j(O$^1$D), which may be a hint of a photo-induced process driving HONO

production. The average peak value of $P_{unknown}$ was comparable to that measured in Beijing (2.58 ppb h$^{-1}$) and is larger

than in Guangzhou (0.65 ppb h$^{-1}$), Santiago (1.77 ppb h$^{-1}$), indicating the important contribution of the unknown source

to HONO production. While HONO concentration is relatively low at noon, the unknown source strength is the largest,

as a strong source is needed to compensate for the rapid photolysis loss of HONO at noon.

Meanwhile, a 0-D box model was employed to understand the observed HONO profiles, and only the default NO+OH

reaction was considered for HONO production (scenario S0 in Table 2). Although the HONO diurnal trend was

captured in the base run simulation, the simulated HONO concentration was about an order of magnitude lower than

the measured concentration (Figure 5), further confirming the crucial role of non-gas-phase sources in HONO

production. The simulated HONO concentration was less than 0.05 ppb around noontime, and the average peak

concentration was only 0.2 ppb.

### 3.3 Correlation analysis

As the unknown source strength of daytime HONO was much greater, we focused on analyzing the unknown source of

daytime HONO in this part. According to our analysis above, photo-induced processes were likely to be the reason for

the high observed HONO during the daytime. Previous studies have proposed some photo-induced heterogeneous

pathways, including photo-induced NO$_2$ to HONO conversion on the aerosol surface, photo-induced NO$_2$ to HONO

conversion on the ground surface, and heterogeneous nitrate photolysis. However, the relative importance of these

sources to HONO production is still highly controversial. The comprehensive measurements performed during the

EXPLORE-YRD campaign provide us with a unique opportunity to evaluate different HONO sources.

The correlation analysis between $P_{unknown}$ and other potential parameters can provide us with a preliminary

understanding of the potential HONO sources. NO$_2$ concentration can be an important indicator of the heterogeneous

reaction of NO$_2$ at the ground surface because the ground surface-to-volume ratio was considered to be constant for a

well-mixed boundary layer. NO$_2\times S_a$ can represent the heterogeneous NO$_2$ reaction on the aerosol surface. The particle



nitrate concentration can be used as an indicator of nitrate photolysis. Therefore, $NO_2 \times j(NO_2)$, $NO_2 \times j(NO_2) \times S_a$, and

$NO_3^- \times j(NO_2)$ can be used as proxies for three heterogeneous HONO sources, respectively. As shown in Figure 6, we found that the unknown source correlated weakly with $NO_2 \times j(NO_2) \times S_a$, indicating photo-induced $NO_2$ conversion on the aerosol surface might play a minor role in daytime HONO production. In contrast, the correlation coefficients ($R^2$) of $P_{unknown}$ with $NO_2 \times j(NO_2)$ and $P_{unknown}$ with $NO_3^- \times j(NO_2)$ ($R^2=0.66$) were significantly higher, suggesting that the photo-induced $NO_2$ conversion on the ground surface and nitrate photolysis may have a profound impact on HONO

production, which will be discussed in more detail in Section 3.4.

### 3.4 HONO formation by different heterogeneous mechanisms

In this section, sensitivity tests were performed to understand how the uncertainties of different photolytic HONO formation pathways would impact HONO production. The goal of these tests was to investigate if the potential HONO formation pathways could explain the profiles of $P_{unknown}$.

**3.4.1 Photo-induced $NO_2$ conversion on the aerosol surface**

The HONO production rate by photo-induced $NO_2$ conversion on the aerosol surface ($P_{aerosol+hv}$) can be expressed as follows:

$$P_{aerosol+hv} = \frac{1}{4}\gamma_{aerosol+hv} \times \frac{j(NO_2)}{0.005 \text{ s}^{-1}} \times [NO_2] \times \upsilon_{NO_2} \times S_a$$

Where $\upsilon_{NO2}$ is the mean molecule speed of $NO_2$, $S_a$ is the aerosol surface-to-volume ratio, and $\gamma_{aerosol+hv}$ is the photo-

induced $NO_2$ uptake coefficient on the aerosol surface which depends on the solar radiation intensity. Previous laboratory studies have reported the positive dependence of $\gamma_{aerosol+hv}$ on solar radiation intensity. Accordingly, $\frac{j(NO_2)}{0.005 \text{ s}^{-1}}$ in the equation represents the promotion effect (Vogel et al., 2003; Wong et al., 2013).

From the equation, we can see that $\gamma_{aerosol+hv}$ is the key factor affecting the HONO production rate. The laboratory-derived $\gamma_{aerosol+hv}$ ranged from $10^{-4}$ to $10^{-7}$, largely depending on the components of the model aerosols used in

laboratory experiments. $\gamma_{aerosol+hv}$ on the soot surface was measured to be $10^{-4}$ (Ammann et al., 1998) but rapidly decreased to $10^{-7}$ due to the consumption of reactive sites (Gerecke et al., 1998; Kleffmann et al., 1999). Stemmler et al. (Stemmler et al., 2007) measured $\gamma_{aerosol+hv}$ on the humic acid aerosol surface up to $2 \times 10^{-5}$, which was considered to be more relevant to ambient conditions. Here we used a photo-induced uptake coefficient of $2 \times 10^{-5}$ and increased the value to $2 \times 10^{-4}$ as an upper limit to check if the photo-induced $NO_2$ conversion on the aerosol surface could explain

$P_{unknown}$. As shown in Figure 7a, HONO production by photo-induced $NO_2$ conversion on the aerosol surface was much lower than $P_{unknown}$ even with a $\gamma_{aerosol+hv}$ upper limit of $2 \times 10^{-4}$, indicating that the contribution of the photo-induced




NO$_2$ conversion on the aerosol surface to HONO is minor, consistent with previous correlation analysis. A recent study also revealed NO$_2$ conversion on the aerosol surface contributed to less than 5% and 12% HONO production in the summer and winter seasons at a rural site, respectively (Song et al., 2022).

It should be mentioned that non-photo-induced NO$_2$ conversion (NO$_2$ hydrolysis) on the aerosol surface also contributed to HONO production during the daytime. However, NO$_2$ hydrolysis is generally considered to be much slower than photo-induced NO$_2$ conversion. Therefore, it couldn't be the main source of daytime HONO in our study and was not discussed here.

### 3.4.2 Heterogeneous nitrate photolysis

The expression for the contribution of heterogeneous nitrate photolysis to HONO production is as follows:

$$P_{NO_3^-+h\nu}=[NO_3^-]\times j(NO_3^-)=[NO_3^-]\times j(HNO_3)_g\times EF$$

Where [NO$_3^-$] is the concentration of nitrate (ppb), $j(HNO_3)_g$ is the photolysis rate constant of gas-phase nitric acid, and EF is the enhancement factor (the ratio of aerosol-phase nitrate photolysis rate constant to gaseous nitrate photolysis rate constant). Recent evidence from a number of studies suggested that aerosol-phase nitrate photolyzes

quickly to generate NO$_2$ or HONO with a rate between 10 and 300 times faster than that of gas-phase HNO$_3$ (Zhou et al., 2011; Ye et al., 2016; Ye et al., 2017). However, there is still great controversy about the rate constant of heterogeneous nitrate photolysis. For instance, observed NO$_x$ and HNO$_3$ during the KORUS-AQ campaign were best explained with a moderate EF of 1-30 by detailed box modeling (Romer et al., 2018). Modeled O$_3$ would be largely overestimated compared to observations if EF was set to 120 in a recent modeling study, indicating EF under

atmospherically relevant conditions may be much lower than 120 (Zhang et al., 2022b). Laufs et al. (Laufs and Kleffmann, 2016) investigated HNO$_3$ photolysis on quartz glass surfaces and found EF value was less than 10. A recent chamber study indicated that suspended nitrate aerosol released HONO with an EF lower than 10 (Shi et al., 2021). Andersen et al. (Andersen et al., 2023) found the EF of marine nitrate aerosol increased with relative humidity and decreased with particulate nitrate concentrations by aircraft and ground measurements, which could partly explain

the large discrepancy in EF values under different conditions.

We performed sensitivity tests with EF values of 20, 50, and 100 to understand how uncertainties in EF would impact HONO production and if particulate nitrate photolysis could explain P$_{unknown}$. As shown in Figure 7b, we calculated the HONO production rate of nitrate photolysis under different EF conditions and found that whether the EF is 20, 50, or 100, the calculated HONO production rate was about one order of magnitude lower than P$_{unknown}$, indicating that the

contribution of nitrate photolysis to HONO is minor at this site due to relatively low concentration of particulate nitrate in summer compared to winter. Zhang et al. (Zhang et al., 2022b) pointed out that HONO production contributed by nitrate photolysis was a magnitude higher than that on clean days.





### 3.4.3 Photo-induced NO$_2$ conversion on the ground surface

Vertical HONO measurements revealed a negative profile (Kleffmann et al., 2003; Wong et al., 2013; Tuite et al.,
2021), providing evidence that a strong HONO source is likely near the ground surface. Laboratory studies confirmed
the speculation and found that photo-induced reactions between NO$_2$ and organic material (e.g., humic acids) lead to
pronounced HONO production (George et al., 2005; Stemmler et al., 2006; Han et al., 2016). Since humic acids are an
important component of soil, and soil constitutes an important portion of the ground surface, this mechanism seems to
be an important candidate for HONO sources in the lower troposphere.

HONO production by photo-induced NO$_2$ conversion on the ground surface can be expressed as follows:

$$P_{ground+h\nu}=\frac{1}{4}\gamma_{ground+h\nu}\times\frac{j(NO_2)}{0.005\ s^{-1}}\times[NO_2]\times\upsilon_{NO_2}\times\frac{1}{MLH}$$

According to laboratory-derived uptake coefficients on humic acids, $\gamma_{ground+h\nu}$ was on the order of $10^{-5}$ (Stemmler et al.,
2006; Stemmler et al., 2007; Han et al., 2016), and sensitivity tests were performed with $\gamma_{ground+h\nu}$ of $2.5\times10^{-5}$ and $4\times10^{-5}$.
In addition, mixing layer height (MLH) was also a crucial factor influencing the HONO production rate on the
ground surface. MLH refers to the height of the atmospheric layer where turbulent mixing occurs and pollutants are
effectively mixed throughout the layer. It is worth mentioning that MLH instead of boundary layer height (BLH) was
used here, as BLH would result in a significant underestimation of surface HONO source strength. Recent HONO
vertical measurements revealed HONO concentration at 100m was evidently lower than that near the surface, implying
MLH was less than 100 m (Vogel et al., 2003; Vandenboer et al., 2013; Xing et al., 2021). Here we take the value of
50 m for the MLH based on previous studies (Lee et al., 2016; Xue et al., 2022; Zhang et al., 2022d). When the
$\gamma_{ground+h\nu}$ was $4\times10^{-5}$, the variation trend and magnitude of HONO production were similar to $P_{unknown}$ (Figure 7c),
indicating photo-induced NO$_2$ conversion on the ground surface was an important HONO source. This was consistent
with some studies combining field measurements and modeling simulations (Wong et al., 2013; Tuite et al., 2021).

### 3.5 HONO formation by vehicle emissions

It is generally accepted that HONO can be emitted by vehicle exhaust, and HONO/NO$_x$ is used as a proxy to quantify
primary HONO emissions from vehicles. Based on previous studies (Kurtenbach et al., 2001; Liu et al., 2019a), we
used a widely-applied ratio of 0.008 to calculate HONO emissions by vehicles at night. Considering the short lifetime
of HONO with respect to photolysis during the daytime, the daytime HONO concentrations contributed by vehicles
would be overestimated if a fixed value of HONO/NO$_x$ is used. Therefore, the atmospheric lifetime of NO$_x$ and HONO
should be taken into consideration to calculate vehicle emissions during the daytime (Xue et al., 2022; Zhang et al.,
2022d).

Primary HONO emissions form vehicles can be expressed as follows:




$$HONO_{emi}=0.008 \times NO_x \quad \text{(nighttime)}$$

$$HONO_{emi}=0.008 \times NO_x \times \frac{\tau(HONO)}{\tau(NO_x)} \quad \text{(daytime)}$$

$\tau(HONO)$ and $\tau(NO_x)$ denotes the atmospheric lifetime of HONO and $NO_x$, respectively. $\tau(HONO)$ can be expressed as follows:

$$\tau(HONO)=\frac{1}{j(HONO)+OH \times k_{OH+HONO}}$$

$NO_x$ in the atmosphere is mainly removed by reaction with OH to generate $HNO_3$ and $N_2O_5$ hydrolysis to generate $HNO_3$. $N_2O_5$ hydrolysis is negligible compared to $NO_2+OH$ during the daytime. Therefore, the atmospheric lifetime of

$NO_x$ can be expressed as follows:

$$\tau(NO_x)=\frac{1}{OH \times k_{OH+NO_2}} \times \left(1+\frac{NO}{NO_2}\right)$$

The average diurnal profile of HONO contributed by vehicle emission is shown in Figure S2. The calculated contribution of vehicle emission to observed HONO ranged from 2% to 32%, with an average value of 15%.

**3.6 Box modeling on HONO production with the incorporation of different HONO sources**

Based on the above analysis, five heterogeneous HONO formation mechanisms were parameterized into the box model. The mechanisms and corresponding parameter settings are listed in Table 1. Such parameter settings are adopted from previous laboratory and modeling studies (Stemmler et al., 2006; Stemmler et al., 2007; Han et al., 2016; Romer et al., 2018; Zhang et al., 2022b).

As shown in Figure 8, included HONO sources significantly improved the model performance in HONO simulations.

The box model incorporated with five heterogeneous sources (S1) well reproduced observed HONO profiles during nighttime, indicating a reasonable representation of nighttime HONO mechanisms. However, HONO concentration was still underestimated from 12:00 LT to 16:00 LT, which was likely caused by the following two reasons: i) the uncertainty of $\gamma_{ground+hv}$, $\gamma_{aerosol+hv}$, and EF; ii) other unknown daytime sources of HONO that were not included in the model. Sensitivity tests were performed to see if the uncertainties of the parameters adopted in models caused the

discrepancy between modeled and observed HONO concentrations during the daytime. The results of each sensitivity run were compared to that of S1, and the description of the changes in each sensitivity test is listed in Table 2. If we increased $\gamma_{ground+hv}$ to $6 \times 10^{-5}$, the HONO concentration from 12:00 LT to 15:00 LT could be explained, whereas the HONO concentration from 6:00 LT-12:00 LT will be largely overestimated (S2 in Figure 9); if we increased $\gamma_{aerosol+hv}$ to $2 \times 10^{-4}$ (S3 in Figure 9), the modeled daytime HONO nearly kept unchanged compared to S1 due to low aerosol

surface to volume ratio; if we increased EF to 100, daytime HONO still couldn't be explained (S4 in Figure 9), indicating underestimation of daytime HONO was not due to the uncertainties of $\gamma_{ground+hv}$, $\gamma_{aerosol+hv}$, and EF but was



more likely due to other sources that were not considered. It should be noted that the box model can't take vertical transport into account, which may introduce additional uncertainty. Wong et al. (Wong et al., 2013) indicated that upward transport played an important role in distributing surface-produced HONO to the entire boundary layer. A 1D

or 3D model may be an ideal tool for simulating HONO. The advantage of using the box model to explore the source of HONO was that the box model was constrained by simultaneously measured HONO precursors, which enabled a more accurate investigation of its secondary sources.

Modeled HONO profiles from May 28 to June 12 are shown in Figure 10. Notably, simulated HONO profiles showed very good agreement with the observations from May 28 to June 4, while underestimating daytime and nighttime

HONO concentrations from June 5 to June 12. Previous studies observed much higher HONO concentrations after fertilization compared to periods before fertilization (Xue et al., 2021). Given that the measurement site was surrounded by agricultural fields and the measurement period covered the fertilization event after the wheat harvest in June, fertilization event-induced soil HONO emissions may result in high daytime HONO levels and the HONO/$NO_x$ ratio. Thereby, field HONO flux measurements and laboratory investigations of corresponding mechanisms are

warranted in future work to more accurately represent soil HONO emissions in models.

### 3.7 HONO budget

Figure 11a depicts the averaged diurnal profiles of the HONO production rates from six HONO sources (five heterogeneous sources and gas-phase NO+OH). During daytime (6:00-19:00), photo-induced $NO_2$ conversion on the ground surface was the main contributor to simulated HONO, accounting for 73% of the total, followed by NO+OH

(10%), $NO_2$ hydrolysis on the ground surface (9%), photo-induced $NO_2$ conversion on the aerosol surface (3%), nitrate photolysis (3%), and $NO_2$ hydrolysis on the aerosol surface (2%). The maximum production rate by photo-induced $NO_2$ conversion on the ground surface was 1.85 ppb h$^{-1}$, which occurred around 9:00 LT. The NO+OH reaction played a minor role in HONO production. This is consistent with recent studies, which also revealed that the NO+OH reaction was too small to explain daytime HONO (Li et al., 2018a; Michoud et al., 2014). In contrast, Liu et al. (Liu et al., 2021)

reported that the NO+OH reaction dominated daytime HONO production in summer in Beijing, contributing 22% of total HONO production. Despite the uncertainties of $\gamma_{aerosol+h\nu}$ and EF, photo-induced $NO_2$ conversion on the aerosol surface and nitrate photolysis contributed to less than 10% of total production, implying they were not important in HONO production at this site. As suggested by previous studies, heterogeneous nitrate photolysis may be more important under low-$NO_x$ conditions (Zhou et al., 2003; Elshorbany et al., 2012). $NO_2$ hydrolysis on the ground

surface dominated HONO production during nighttime, contributing to 65% of total production, consistent with previous studies (Kleffmann et al., 2003; Vandenboer et al., 2013). HONO loss was dominated by photolysis, followed by deposition and reaction with OH (Figure 11b). The average peak loss rate by HONO photolysis was 2.3 ppb h$^{-1}$.





## 3.8 HONO contribution to $RO_x$ and $O_3$ production

The primary sources of $RO_x$ radicals mainly include $O_3$ photolysis, HONO photolysis, ozonolysis of alkenes, and
carbonyl photolysis. Generally, in clean regions, $O_3$ photolysis is the main contributor to the primary source of radicals,
while in polluted urban areas, HONO photolysis would exceed $O_3$ photolysis and become the main primary radical
source. Field measurements revealed that HONO photolysis contributed to more than 40% of primary radical sources
(Tan et al., 2017; Ma et al., 2019; Ren et al., 2006). Ozonolysis of alkenes is an important source of $RO_x$ radicals at
night. In addition, ozonolysis of alkenes was the second largest radical source during the winter in urban areas (Tan et
al., 2018). During the winter in oil- and natural gas-producing basins in the United States, carbonyl photolysis was
found to be the largest radical source, initiating the photochemical chain reactions and ultimately leading to
exceedances of $O_3$ concentration (Edwards et al., 2014). Similarly, carbonyl photolysis was the major primary source
of radicals in the oil extraction region in China during the winter seasons (Chen et al., 2020).

Figure 12 depicts the contribution of different channels to the primary source of $RO_x$ in the Taizhou area. We can
clearly see that HONO photolysis is the dominant contributor to the primary source of $RO_x$, not only in the early
morning but also throughout the day. HONO photolysis contributed to 43% of $RO_x$ production during the daytime,
followed by $O_3$ photolysis (17%), HCHO photolysis (14%), ozonolysis of alkenes (12%), and carbonyl photolysis
(10%). The maximum $RO_x$ production rate by HONO photolysis was 2.6 ppb $h^{-1}$, which occurs around 11:00 LT.
HONO photolysis was also identified as the major source of primary $RO_x$ production in the summer of Chengdu (34%),
and Wangdu (38%) (Tan et al., 2017; Yang et al., 2021b). The results above indicate that HONO is closely linked to
the $RO_x$ budget, and it is crucial to accurately identify HONO sources in order to fully comprehend the atmospheric
oxidation capacity.

As mentioned in Section 3.1, $O_3$ episodes were typically accompanied by high HONO levels. In this section, a box
model was employed to quantify the contribution of HONO to $O_3$ production. $F(O_3)$ and $P(O_3)$ was first calculated
with observed HONO as constraint. $F(O_3)$ showed a typical diurnal pattern with a maximum of 15.8 ppb $h^{-1}$ occurring
around 9:00 LT (Figure 13). When observed HONO was not constrained in the model, the average peak of $F(O_3)$
decreased nearly 45% to 8.8 ppb $h^{-1}$ (Figure 13), indicating HONO significantly contributed to $O_3$ production and
thereby boosted $O_3$ pollution during the summer. $P(O_3)$ decreased from 12.6 ppb $h^{-1}$ to 6.7 ppb $h^{-1}$ with a 47%
reduction, which is higher than the 20% reduction in urban Beijing (Li et al., 2021; Zhang et al., 2023). Yang et al.
(Yang et al., 2021c) observed two $O_3$ pollution episodes with nearly identical $O_3$ precursor (VOC and $NO_x$) levels.
Interestingly, they found higher $O_3$ peak concentrations coinciding with higher HONO concentrations in one of the
episodes. This correlation can be attributed to the fact that higher HONO concentrations can lead to higher levels of
OH radicals, which can, in turn, enhance the production of $O_3$ by initiating the chain reactions (Yang et al., 2021c). We
also modeled OH radicals with and without observed HONO as a model constraint. The modeled OH averaged peak



concentration with HONO was $8\times10^6$ cm$^{-3}$, which was nearly twice as high as that ($4.4\times10^6$ cm$^{-3}$) without HONO as a model constraint. Our results emphasized the importance of HONO in primary radical production and $O_3$ production. Accurately representing HONO sources in models is crucial for characterizing atmospheric oxidation capacity and quantifying secondary pollutants production. Therefore, more field, laboratory, and modeling studies are urgently needed to develop more concise HONO formation parameterizations. In addition, more and more studies are focusing

on VOCs and $NO_x$ reduction to achieve $O_3$ mitigation. Our results suggested HONO contributed significantly to $O_3$ production in China, and thereby, reducing HONO production may be an alternative way for $O_3$ control. As $NO_2$ heterogeneous reactions on the ground surface were important sources for HONO production, reducing $NO_x$ emissions would be beneficial for reducing HONO emissions. However, $NO_x$ reduction may also lead to more $O_3$ production if $O_3$ formation is in a VOC-limited regime, and hence the overall effects of $NO_x$ reduction on $O_3$ should be evaluated by

chemical transport models.

## 4. Conclusions

HONO measurements, along with a wide range of gas-phase, aerosol-phase, and meteorological measurements, were conducted during the EXPLORE-YRD campaign. HONO concentrations exhibited significant day-to-day variations, with the highest concentration reaching 4.4 ppb and the lowest concentration of several tens of ppt. The relatively

higher summer HONO and OH concentrations compared to other suburban sites suggested higher atmospheric oxidation capacity. HONO/$NO_x$ exhibited a diurnal pattern similar to $j(O^1D)$, with a maximum of 0.17 around noon, implying the existence of photo-induced HONO sources. The box model with the only default HONO source (OH+NO) failed to capture the observed HONO concentration. The calculated $P_{unknown}$ showed a diurnal pattern with a maximum of 2.5 ppb h$^{-1}$ occurring around 10:00-12:00 LT. The correlation coefficients of $P_{unknown}$ with $NO_2\times j(NO_2)$ and $NO_3^-$

$\times j(NO_2)$ were significantly higher than that of $P_{unknown}$ with $NO_2\times j(NO_2)\times S_a$, indicating photo-induced $NO_2$ conversion on the aerosol surface played a minor role in HONO production. Sensitivity tests suggested HONO production rate by photo-induced $NO_2$ conversion on the ground surface was able to explain $P_{unknown}$. With additional HONO sources incorporated, the box well reproduced the nighttime HONO concentration but still underestimated the daytime HONO concentration. Further sensitivity tests implied the discrepancy between modeled and observed daytime HONO

concentration was not due to uncertainties in $\gamma_{ground+hv}$, $\gamma_{aerosol+hv}$, and EF but was more likely due to other HONO sources that were not considered. Among the heterogeneous and gas-phase HONO sources, photo-induced $NO_2$ conversion on the ground surface was the main contributor to simulated daytime HONO, accounting for 73% of the total, followed by NO+OH (10%), $NO_2$ hydrolysis on the ground surface (9%), photo-induced $NO_2$ conversion on the aerosol surface (3%), nitrate photolysis (3%), and $NO_2$ hydrolysis on the aerosol surface (2%). The maximum




production rate of photo-induced $NO_2$ conversion on the ground surface was 1.85 ppb h$^{-1}$ which occurred around 9:00 LT. $NO_2$ hydrolysis on the ground surface was the major source of nighttime HONO, contributing to 65% of total HONO production. HONO loss was dominated by photolysis, followed by deposition and reaction with OH.

HONO photolysis contributed to 43% of $RO_x$ production during the daytime, followed by $O_3$ photolysis (17%), HCHO photolysis (14%), ozonolysis of alkenes (12%), and carbonyl photolysis (10%). The maximum $RO_x$ production rate by

HONO photolysis was 2.6 ppb h$^{-1}$, which occurs around 11:00 LT. When the observed HONO was constrained in the box model, the calculated $P(O_3)$ showed a typical diurnal pattern with a maximum of 12.6 ppb h$^{-1}$ occurring around 9:00 LT. $P(O_3)$ would decrease by 45% to 6.7 ppb h$^{-1}$ if HONO was not constrained, indicating HONO significantly enhanced $O_3$ production and hence aggravated $O_3$ pollution during the summer season.

**Data availability.** The data used in this study are available from the corresponding author upon request (k.lu@pku.edu.cn).

**Author contributions.** KL and YZ designed the experiments. CY analyzed the data and wrote the manuscript with input from all the authors.

**Competing interests.** The contact author has declared that neither they nor their co-authors have any competing
interests.

**Acknowledgements.** We thank the science teams of the EXPLORE-YRD campaign for their support.

**Financial support.** This work was supported by the National Natural Science Foundation of China (grant nos. 21976006) and the China Postdoctoral Science Foundation (grant nos. 2022T150011).





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



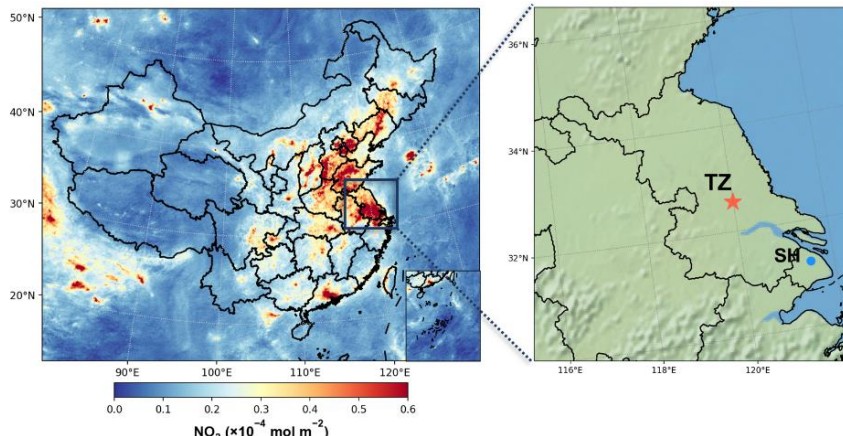

**Figure 1: Location of the field measurement site (red star) in Taizhou (TZ), Jiangsu Province. This site is situated approximately 200 km northwest of Shanghai (SH), one megacity in YRD. The left map is colored by monthly average NO₂ column density retrieved from TROPOMI (July, 2018).**



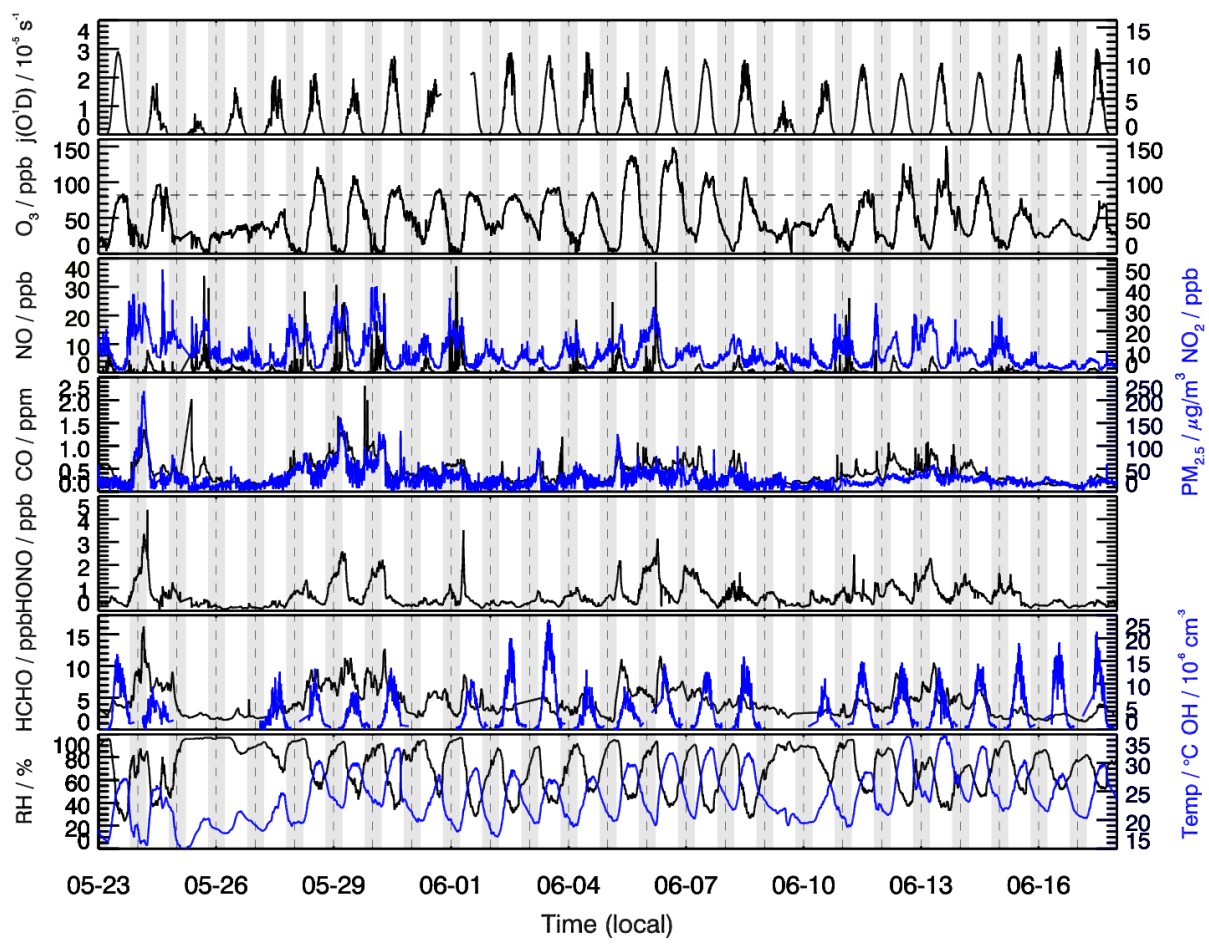

**Figure 2: Time series of HONO, O₃, CO, PM₂.₅, OH, HCHO, NOₓ, relative humidity (RH), temperature and j(O¹D) during the EXPLORE-YRD campaign.**








**Figure 3: The diurnal patterns of HONO, HONO/NOₓ, NO, NO₂, CO, O₃, HCHO and j(O¹D).**




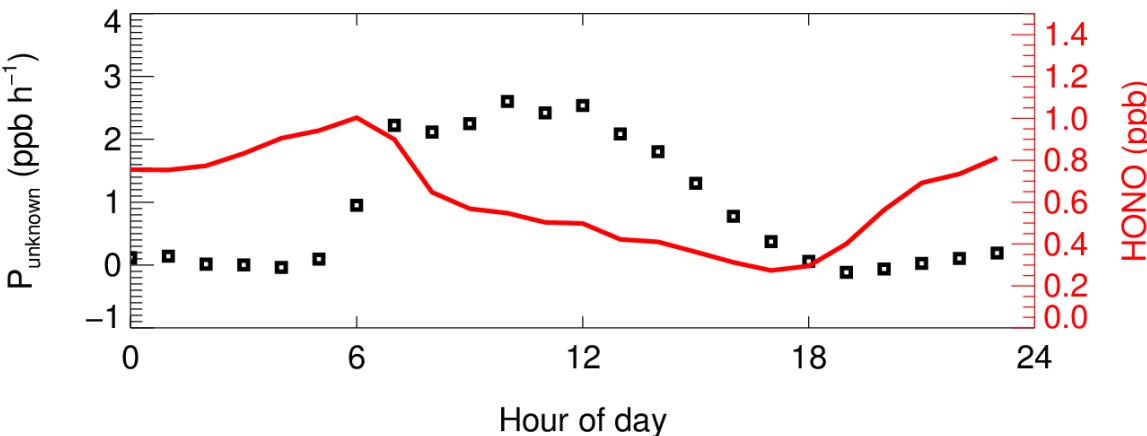

**Figure 4: The diurnal profiles of calculated HONO unknown source ($P_{unknown}$) strength and observed HONO.**








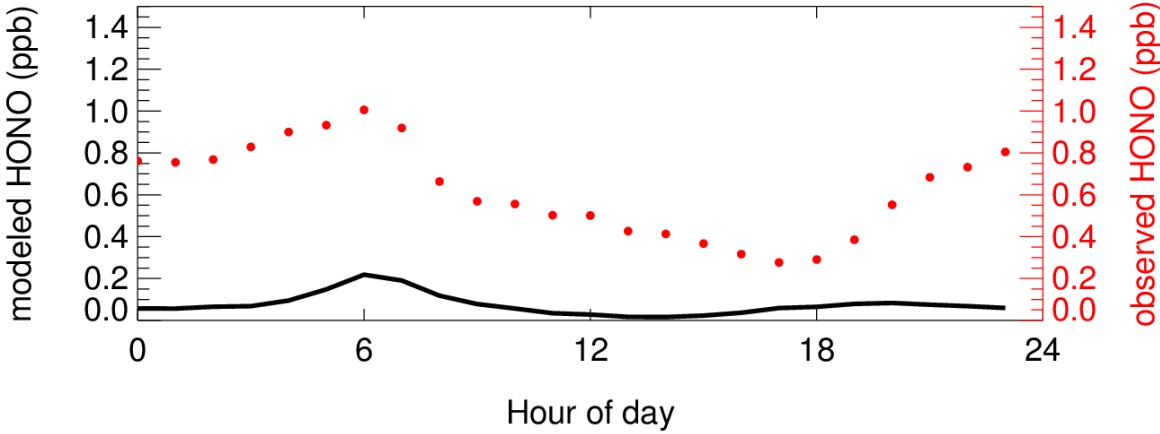

**Figure 5: The modeled HONO concentration with only default HONO source (OH+NO).**










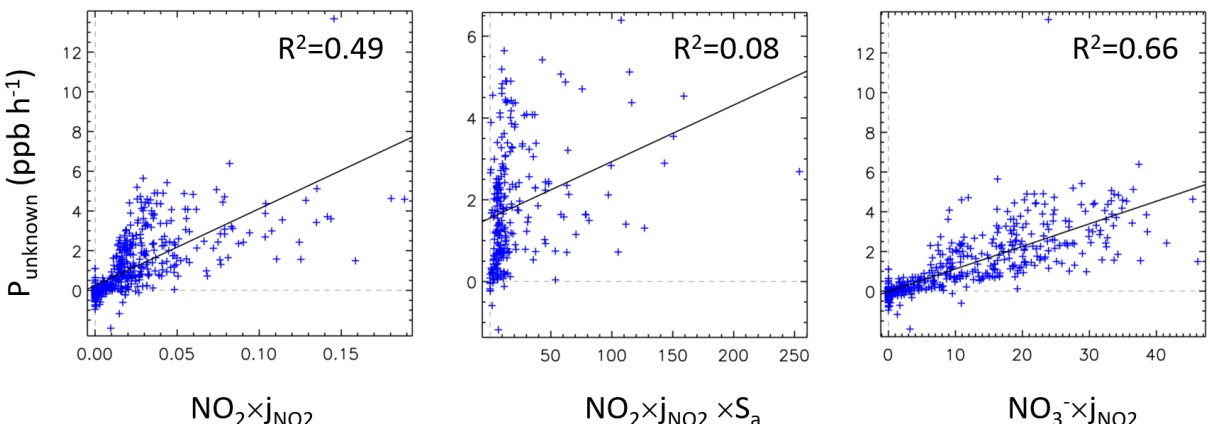

**Figure 6: The unknown source strength ($P_{unknown}$, 8:00-18:00) plotted against $NO_2 \times j(NO_2)$, $NO_2 \times j(NO_2) \times S_a$ and $NO_3^- \times j(NO_2)$.**







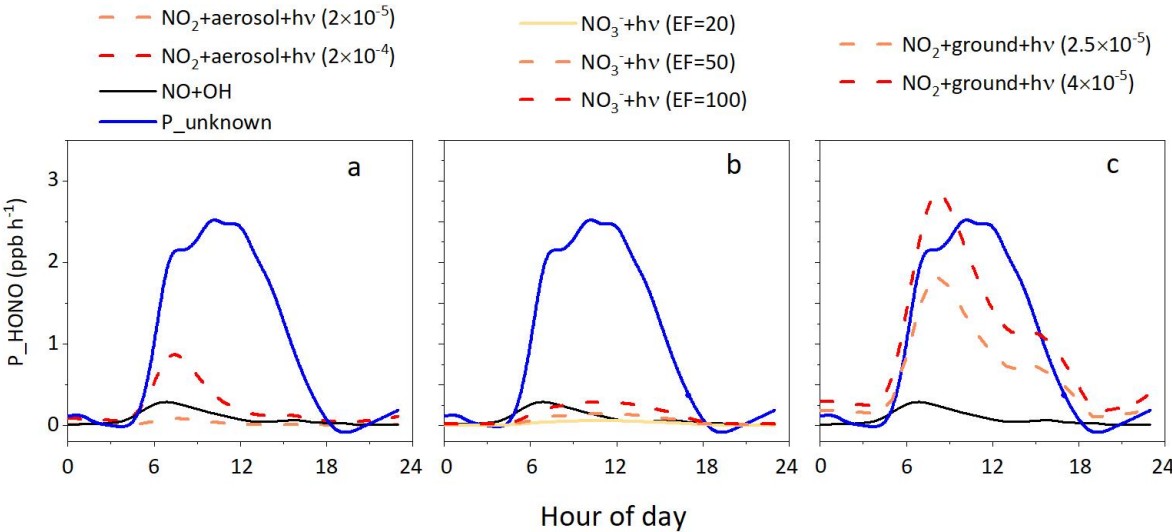

**Figure 7: HONO production rates by different HONO formation pathways with different parameter settings. (a) photo-induced NO₂ conversion on the aerosol surface; (b) heterogeneous particulate nitrate photolysis; (c) photo-induced NO₂ conversion on the ground surface.**





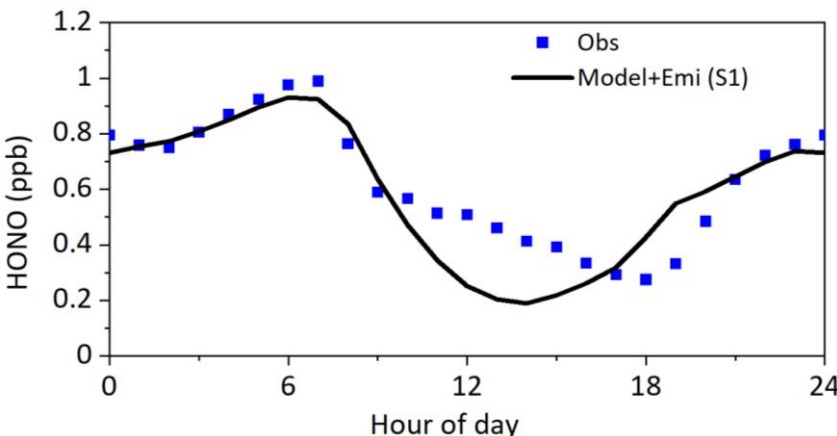

**Figure 8: Modeled HONO concentrations with additional HONO sources compared to the observations. Model+Emi represents the sum of modeled HONO and vehicle HONO emission.**








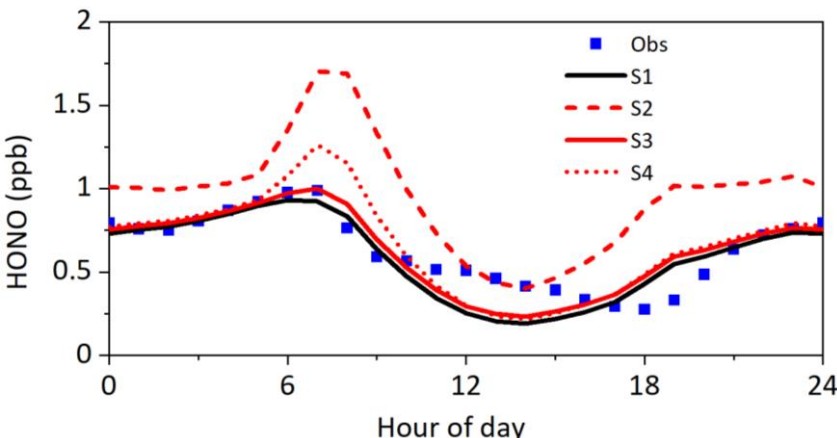

**Figure 9: Results from sensitivity tests compared with observed HONO concentrations. S1: parameters were listed in Table 1; S2: $\gamma_{ground+h\nu}$ was changed to $6\times10^{-5}$ compared to S1; S3: $\gamma_{aerosol+h\nu}$ was changed to $2\times10^{-4}$ compared to S1; S4: EF was changed to 120 compared to S1.**



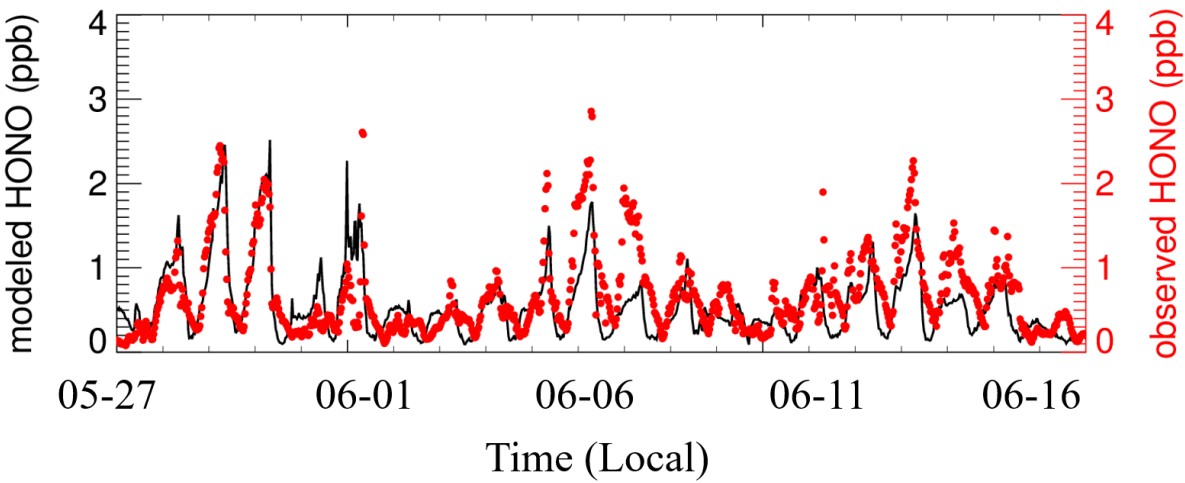

**Figure 10: Time series of modeled and observed HONO concentrations from 05-27 to 06-16.**








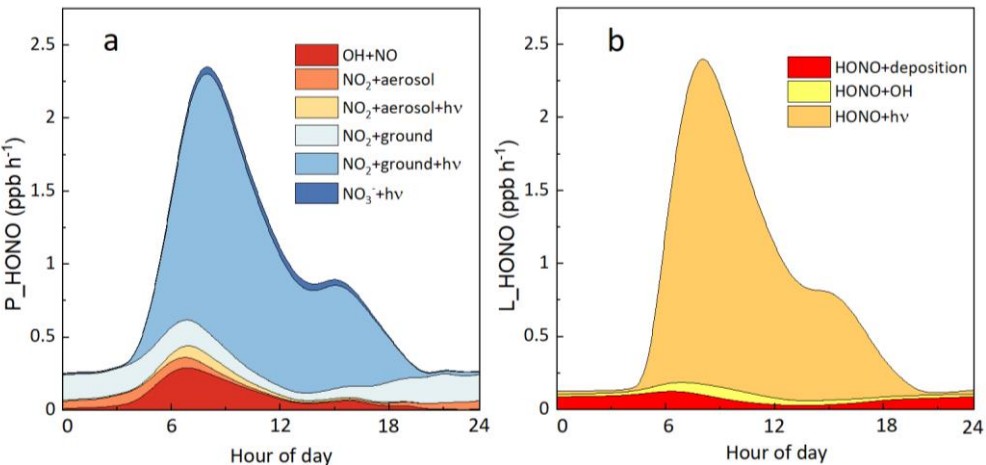

**Figure 11: HONO production rates and loss rates by different pathways.**









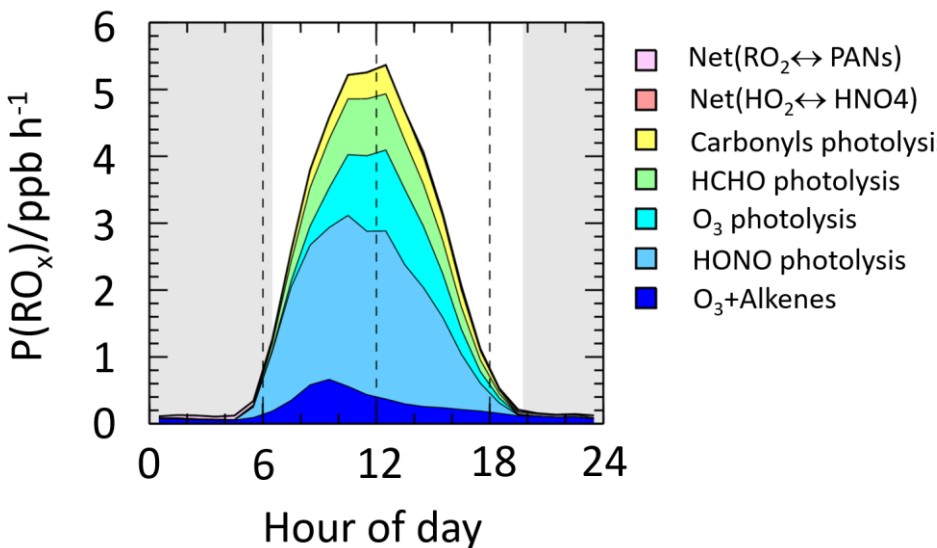

**Figure 12: Primary RO$_x$ production rates by different pathways during the EXPLORE-YRD campaign.**





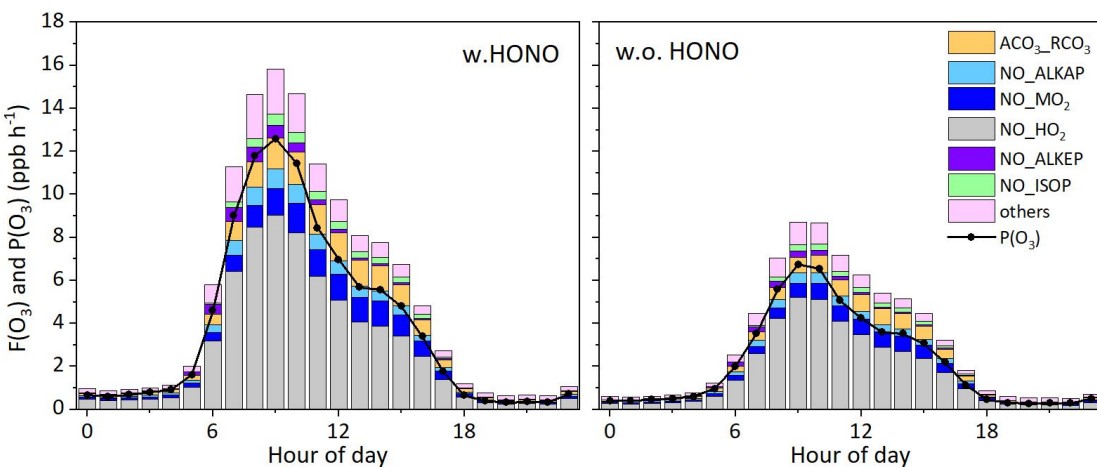

**Figure 13: Model-calculated total and net O₃ production rate with and without observed HONO as a model constraint.**








**Table 1. Parameterized HONO source mechanisms in the box model.**

| Mechanisms | Reaction | Parameterization |
|---|---|---|
| NO$_2$ hydrolysis on the aerosol surface | NO$_2$+aerosol→0.5HONO | $\gamma_{aerosol}$=1×10$^{-5}$ |
| Photo-induced NO$_2$ conversion on the aerosol surface | NO$_2$+aerosol $\xrightarrow{hv}$ HONO | $\gamma_{aerosol+hv}$=2×10$^{-5}$ |
| NO$_2$ hydrolysis on the ground surface | NO$_2$+ground→0.5HONO | $\gamma_{ground}$=2×10$^{-6}$ |
| Photo-induced NO$_2$ conversion on the ground surface | NO$_2$+ground $\xrightarrow{hv}$ HONO | $\gamma_{ground+hv}$=2.5×10$^{-5}$ |
| Nitrate photolysis | NO$_3^-$ $\xrightarrow{hv}$ HONO | EF=20 |







**Table 2. Different model simulation scenarios and corresponding configuration.**

| Scenario | Configuration |
|----------|---------------|
| S0 | Default HONO formation mechanism (OH+NO) |
| S1 | S0+five heterogeneous source (parameters set as Table 1) +emission |
| S2 | $\gamma_{ground+hv}$ was changed to $6\times10^{-5}$ compared to S1 |
| S3 | $\gamma_{aerosol+hv}$ was changed to $2\times10^{-4}$ compared to S1 |
| S4 | EF was changed to 120 compared to S1 |
