# Peer review of "HONO chemistry at a suburban site during the EXPLORE-YRD campaign in 2018: formation mechanisms and impacts on O3 production"

_EGUsphere, 2023_

## Author Comment (AC1)

Response to referee 1:

We thank the anonymous referee 1 for the supportive comments and suggestions that have improved the clarity of the manuscript. Please find below a detailed response to each suggestion. Comments by the reviewer are given in black normal font, and our response to the comments is shown in blue. Newly added and modified text in the revised manuscript and supporting information (SI) is given in italics.

This paper reports observations of HONO in a suburban location in the YRD region over a period of several weeks in summer. The authors find that photo-induced NO2 conversion on the ground dominated the HONO production during the daytime, and NO2 hydrolysis on the ground surface was the major source of nighttime HONO. Meanwhile, the authors employ a box model to investigate what contribution ROx and O3 derived from HONO makes to the radical chemistry at their measurement site. These results are meaningful for the development of HONO investigation. However, there also existed some problems the authors need to improve the manuscript before its publication in ACP.

1. I am curious about the observation time. In the Method section, the observation period is introduced to be from May 14 to June 20, 2018, but Figure 2 only presents the observed parameters from May 23 to June 18, 2018, whereas the box model simulates the period of May 28-June 12, 2018. Why?

The campaign took place from May 14, 2018, but some instruments were not in good state at the beginning. Considering the availability of complete data, we focus on the period of May 23 to June 18. As suggested, we revised the figure and showed the modeled and measured HONO profiles from May 23 to June 18.

[Figure]

*Figure 9: Time series of modeled and observed HONO concentrations from 05-27 to 06-18.*

2. Importantly, in the calculation of HONO unknown source strength, the HONO deposition was not considered, why?

HONO deposition loss can be parametrized by multiplying the measured HONO concentration with the dry deposition velocity and then scaling by the boundary height.

If we take a HONO deposition velocity of 2 cm s$^{-1}$ and a boundary height of 1000 m, HONO loss by deposition is in the order of a few ppt h$^{-1}$ in our study which is indeed small (<4 % of HONO loss by photolysis during 10:00–14:00 LT) compared to HONO loss with respect to photolysis. Therefore, for simplification, HONO loss by deposition was not considered by our study and also in some other studies when calculating the unknown source strength (Sörgel et al.,2011; Xue et al., 2022).

3. In the section of vehicle emission, the calculated average contribution of vehicle emission to observed HONO could reach 15%, but it did not appear in the HONO budget, why? According to the HONO budget result, direct emission might be the second most important source for HONO.

As suggested, we have taken vehicle HONO emission into consideration. The revised figure was shown below. HONO production by vehicle emission accounted for 22% of the seven HONO sources during nighttime, while it played a minor role during the daytime.

[Figure]

*Figure 10: HONO production rates and loss rates by different pathways.*

Line 2: The secondary HONO should be removed in the title.

We have removed the second "HONO" in the title.

Line 43: SOA should be presented as its full name when it appeared at the first time.

As suggested, we have changed SOA to be secondary organic aerosol (SOA).

Line 47: …HONO was a vital OH precursor not only in the early morning but also throughout the day.

The corresponding sentence has changed as follows:
Line 48-49:

*"Previous studies reported that HONO was a vital OH precursor not only in the early morning but also throughout the day."*

Line 52/87: varied – various

Changed accordingly.

Line 53: remove "to explain HONO"

Changed accordingly.

Line 76: remove "typically"

We have removed "typically".

Line 80: heterogeneous nitrate/HNO3 photolysis on varied surfaces – adsorbed nitrate/HNO3 photolysis

The corresponding texts have been changed as follows:
Line 81-82:
*"In addition, adsorbed nitrate/HNO$_3$ photolysis on various surfaces was found to be enhanced compared to gas-phase HNO$_3$ and also contributed to HONO formation."*

Line 82: heterogeneous – adsorbed

Changed accordingly.

Line 98/258/321/345/348/356/418/439/474: write the right format (e.g., Fu et al., (2019) found…) for the references.

Thank the reviewer for noticing the errors. We have corrected accordingly:
"Fu et al. (Fu et al., 2019)" to Fu et al. (2019)
"Liu et al. (Liu et al., 2019a)" to "Liu et al. (2019a)"
"Stemmler et al. (Stemmler et al., 2007)" to "Stemmler et al. (2007)"
"Laufs et al. (Laufs and Kleffmann, 2016)" to "Laufs et al. (2016)"
"Andersen et al. (Andersen et al., 2023)" to "Andersen et al. (2023)"
"Zhang et al. (Zhang et al., 2022b)" to "Zhang et al. (2022b)"
"Wong et al. (Wong et al., 2013)" to "Wong et al. (2013)"
"Liu et al. (Liu et al., 2021)" to "Liu et al. (2021)"
"Yang et al. (Yang et al., 2021c)" to "Yang et al. (2021c)"

Line 150: throughout the paper – in this study

Changed accordingly.

Line 155: an instrument model is needed for the portable weather station.

MAWS301, Vaisala, Finland

Line 162: try – trying

Changed accordingly.

Line 180: the reaction of NO2 and OH is missing in D(Ox)

Yes, we have considered $NO_2+OH$ reaction in calculating $O_3$ loss rate, but missed it in the expression. The expression has corrected as follows:
Line 181-182:

$$D(O_x)=k_{O^1D+H_2O}[O^1D][H_2O]+[O_3]\left(k_{O_3+Alkenes}[Alkenes]+k_{O_3+HO_2}[HO_2]+k_{O_3+OH}[OH]\right)+k_{NO_2+OH}[OH][NO_2]+$$

$$3\left(k_{O_3+NO_2}[NO_2][O_3]-k_{NO+NO_3}[NO_3][NO]-j_{NO_3}[NO_3]\right)$$

Line 186/348: relative humidity has been abbreviated in line 154.

We have changed "relative humidity" to "RH".

Line 191: it is difficult to derive the deduction of "VOCs are abundant" from "the MAXIMUM diurnal averaged HCHO concentration". Please rephrase the sentence.

We have rephased the texts as follows:
Line 192-193:
*Similar with CO, HCHO peaked around 8:00 LT with a maximum of 5 ppb, indicating the effect of anthropogenic emission-related sources.*

Line 197: the Class-II limit values are corresponding to the maximum 8-hour averaged O3, rather than O3 concentration.

We revised the corresponding texts as follows:
Line 196-199:
*"The daily maximum 8-hour average $O_3$ concentrations throughout the observation period frequently exceeded Class-II limit values (160 μg m$^{-3}$, which is equivalent to 82 ppb at 298 K and 1013 kpa) of the National Ambient Air Quality Standard, and the highest $O_3$ concentration can reach as high as 150 ppb, indicating serious photochemical pollution."*

Line 207: What does mean the average peak concentration of OH? For example, in the study of Zhang et al., (2022a), the OH concentration of 2.7*10^6 cm-3 represented the

average OH radical concentration at noontime (11:00-13:00). Comparison should be performed at the same level.

The reviewer is right. The average peak concentration means the peak OH concentration shown in the averaged diurnal profile, which typically occurred around noon. We have removed the citation of Zhang et al., (2022a).

Line 220: was possibly the reason – was the possible reason

Changed accordingly.

Line 227: need the reference for the HONO lifetime. Generally, nocturnal HONO lifetime is relatively long (several hours).

Sorry for the misleading. We mean the lifetime of HONO around noon. We have rephased the texts and add reference.
Line 225-228:
*"Considering that the atmospheric lifetime of HONO is only 10-20 min around noon (with respect to photolysis) (Sörgel et al., 2011), however, the averaged noon-time HONO concentration was relatively high (0.5 ppb), which implied the existence of strong daytime HONO sources to counteract its rapid photolysis."*

Line 235: higher concentration of O3 production – higher O3 production

Changed accordingly.

Line 255: Jinan – Ji'nan

Changed accordingly.

Line 307: photolytic – photo-related?

Yes

Figure 2: the order of magnitude for OH is not 10^(-6) but 10^6.

We have revised Figure 2.

[Figure]

*Figure 1: Time series of HONO, O₃, CO, PM₂.₅, OH, HCHO, NOₓ, relative humidity (RH), temperature and j(O¹D) during the EXPLORE-YRD campaign.*

Figure 13: the meaning of legend should be stated one by one.

As suggested, the legend of Figure 13 has revised as follows:
Line

*Figure 12. Model-calculated total and net O₃ production rate with and without observed HONO as a model constraint. According to different type of VOC precursors, organic peroxy radicals (RO₂) can be classified into seven categories, including methyl peroxy radicals (MO₂=CH₃O₂), first-generation peroxy radicals from alkanes (ALKAP), alkenes except isoprene (ALKEP), isoprene (ISOP), aromatics (AROP), OVOC (OVOCP) and acyl peroxy radicals (ACETYLP=ACO₃+RCO₃).*

Reference:
Sorgel, M., Regelin, E., Bozem, H., Diesch, J. M., Drewnick, F., Fischer, H., Harder, H., Held, A., Hosaynali-Beygi, Z., Martinez, M., and Zetzsch, C.: Quantification of the unknown HONO daytime source and its relation to NO₂, Atmos Chem Phys, 11, 10433-10447, 10.5194/acp-11-10433-2011, 2011.

Xue, C., Ye, C., Kleffmann, J., Zhang, W., He, X., Liu, P., Zhang, C., Zhao, X., Liu, C.,

Ma, Z., Liu, J., Wang, J., Lu, K., Catoire, V., Mellouki, A., and Mu, Y.: Atmospheric measurements at Mt. Tai – Part II: HONO budget and radical ($RO_x$+$NO_3$) chemistry in the lower boundary layer, Atmos. Chem. Phys., 22, 1035-1057, 10.5194/acp-22-1035-2022, 2022.

---

## Author Comment (AC2)

Response to referee 2:

The authors would like to thank the anonymous referee 2 for taking the time to review the manuscript. We thank them for their kind and encouraging words, and for the very relevant comments that allowed us to improve the manuscript. Comments by the reviewer are given in black normal font, and our response to the comments is shown in blue. Newly added and modified text in the revised manuscript and supporting information (SI) is given in italics.

The manuscript examined the potential formation pathways for HONO and their impacts on ozone production in a suburban site in the China YRD region during the summertime, using sophisticated field measurements and constrained box model tests. They found that the traditional OH+NO pathway only largely underestimate the observed HONO concentrations. Within several potential pathways, photo-induced NO2 conversion on the ground is mostly likely the missing HONO source during the day, and NO2 hydrolysis on the group surface is the major missing source at night. The study also indicated a significant HONO contribution from direct vehicle emissions. They also assessed the contributions of the missing sources to the ozone production, suggesting an important role of HONO in aggravating ozone pollution. The topic is important and relevant. The dataset and analysis are comprehensive and valuable in improving the understanding the secondary pollutions and control policies. This paper is within the scope of ACP and might be of great interest to the broad atmospheric science community. I have a few questions and comments that should be answered before it can be considered for publication.

Thank you for your positive comments.

Specific comments:

Line 36: Should it be "increased by 88%"? (12.6-6.7)/6.7 = 88%

We agree with the reviewer. We have changed the texts accordingly:
Line 36-39:
*"The net ozone production rate (6.7 ppb $h^{-1}$) without observed HONO as a model constraint increased by 88% compared to that (12.6 ppb $h^{-1}$) with HONO as a model constraint, indicating HONO evidently enhanced HONO production and hence aggravated $O_3$ pollution in summer seasons."*

Line 64: "which is typically less than 2% NOx emissions" is confusing. Did you mean the HONO/NOx ratio is typically less than 2%?

Yes, we mean that HONO/$NO_x$ ratio is typically less than 2%.

Section 3.7 HONO Budget: why did you not include direct emissions, such as vehicle emission, into the HONO production rate? It seems vehicle emission contributed

significantly to this site (~15%).

As suggested, we have taken vehicle HONO emission into consideration. The revised figure was shown below. HONO production by vehicle emission accounted for 22% of the seven HONO sources during nighttime, while it played a minor role during the daytime.

[Figure]

*Figure 10: HONO production rates and loss rates by different pathways.*

Technical corrections:

Line 28: change it to be "more likely due to".

We have revised accordingly.

Line 37: Should it be "indicating HONO evidently enhanced O3 production"?

Thank you for noticing this mistake. We have changed the corresponding texts as follows:
Line 38:
*"...with HONO as a model constraint, indicating HONO evidently enhanced $O_3$ production".*

Line 42: please define "SOA", also "VOCs", "PAN" …in the following text.

Thank you for your suggestion. We have changed "SOA" to "secondary organic aerosol (SOA)", "VOCs" to "volatile organic compounds (VOCs)", "PAN" to "peroxyacetyl nitrate (PAN)".

Line 52-53: may change the sentence to be "several HONO sources, including …, have been proposed".

We have changed the texts as suggested:

Line 52-53:
*"Till now, several HONO sources, including gas-phase reactions, direct emissions…"*

Line 52, 80, 87: replace "varied" with "various".

"Varied" replaced with "various", as suggested.

Line 98: please correct the format of the citation.

We have corrected "Fu et al. (Fu et al., 2019)" to "Fu et al. (2019)".

Line 105: from 28 to 76 is more than doubling.

The reviewer is correct. We have changed the corresponding texts as follows:
Line 105-106:
*"Recently, this area has witnessed an evident increase in $O_3$ levels, with $O_3$ pollution days more than doubling (28 days to 76 days) from 2014 to 2017 (Liu et al., 2020)."*

Line 314: Should "Sa" here be the "aerosol surface area density", rather than "aerosol surface-to-volume ratio"?

Yes, "Sa" represents the "aerosol surface area density". We have corrected in the revised manuscript accordingly.

Line 345: please correct the format of the citation.

We have changed "Laufs et al. (Laufs and Kleffmann, 2016)" to "Laufs et al. (2016)."

Line 353: change "whether" to be "regardless of whether".

We have changed accordingly.

Figure 2: Some values on y-axis of CO, PM2.5, and NO2 overlaps with each other. Please fix that.

As suggested, we have revised the figure as follows:

[Figure]

*Figure 1: Time series of HONO, O₃, CO, PM₂.₅, OH, HCHO, NOₓ, relative humidity (RH), temperature and j(O¹D) during the EXPLORE-YRD campaign.*

Table 1: Why the Reaction of NO2 hydrolysis only gives 0.5 HONO for 1 NO2 reacted?

$NO_2$ hydrolysis reaction proceeds as follows:

$$2NO_2+H_2O \rightarrow HNO_3+HONO$$

Therefore, one $NO_2$ molecule will lead one 0.5 $HNO_3$ and 0.5 HONO.

---

## Author Comment (AC3)

Response to referee 3:
The authors would like to thank the anonymous referee 3 for taking the time to review the manuscript. We thank for validating our work and for providing us with valuable insights that allowed us to improve the manuscript. Comments by the reviewer are given in black normal font, and our response to the comments is shown in blue. Newly added and modified text in the revised manuscript and supporting information (SI) is given in italics.

Ye et al. examined HONO chemistry and its impact on ozone formation using a field campaign measurement and box modeling. They found a high HONO/NOx ratio of 0.17 around noon coinciding with high J(O1D), which suggests the importance of photo-induced sources for HONO formation. This is furthered verified by statistical analysis and box modeling with updated parameterization. They also demonstrated HONO chemistry can greatly enhance net ozone production by 45%.

Overall, this is a well-executed study and the key conclusions are reasonably defended. In particularly, the observational constraint for HONO chemistry from EXPLORE-YRD campaign adds important evidence to the understanding of HONO formation. However, I suggest the authors to discuss more broadly the HONO formation chemistry under different chemical conditions. I would recommend its publication after revision.

Thank you for your positive comments on this study.

We have expanded some discussion on HONO formation chemistry under different chemical conditions:

Line 446-454:

*"Despite a good match between observed and measured HONO during most days, HONO was still underestimated after fertilization events, indicating the strong influence of soil HONO emission on HONO budget in areas surrounded by agricultural fields. Therefore, soil HONO emission should be well constrained, especially for rural areas during fertilization periods. In addition, while nitrate photolysis played a negligible role in HONO formation in our study, it may play a more important role in winter polluted periods with high nitrate loadings. Our study highlighted important role of $NO_2$ conversion on ground surface. Previous studies found some coexisted gas species like $NH_3$, $CO_2$ may promote HONO production by $NO_2$ heterogeneous reaction (Li et al., 2018; Xu et al., 2019; Liu et al., 2023; Xia et al., 2021). Thereby, laboratory experiments investigating heterogeneous $NO_2$ conversion on ground surface with these species present are needed for better representation of HONO formation in models."*

The authors highlighted HONO production from NO2 heterogeneous conversion at ground. But how is daytime PM2.5 concentration during EXPLORE-YRD? I am wondering if the importance of aerosol update will increase over a severe PM2.5 pollution episode. Some discussion on the application of key conclusion from this study is required.

The times series of $PM_{2.5}$ was shown in Figure 1 in the revised manuscript. We can

clearly see that the maximum PM$_{2.5}$ concentration during the daytime was below 100 µg m$^{-3}$. HONO production by NO$_2$ uptake on aerosol surface can be expressed as follows:

$$P_{aerosol+h\nu}=\frac{1}{4}\gamma_{aerosol+h\nu}\times\frac{j(NO_2)}{0.005\ s^{-1}}\times[NO_2]\times\upsilon_{NO_2}\times S_a$$

If we take an $\gamma_{aerosol+h\nu}$ of $2\times10^{-5}$ and double daytime PM$_{2.5}$ concentrations (assuming Sa was linearly correlated with PM$_{2.5}$ concentrations), then the HONO production by NO$_2$ uptake on aerosol surface was shown below (Figure R1):

[Figure]

Figure R1. HONO production rates by photo-induced NO$_2$ conversion on the aerosol surface.

Despite doubling daytime PM$_{2.5}$ concentrations, HONO production was still much lower than unknown sources, and also lower than NO+OH, implying NO$_2$ uptake on aerosol surface was not important in our study. Compared to NO$_2$ conversion on ground surface, NO$_2$ conversion on aerosol surface was minor, which was ascribed to much smaller surface area to volume ratio ($<0.01$ m$^{-1}$ vs $0.3$ m$^{-1}$)

Our study provides three important hints: 1) NO$_2$ heterogeneous on ground surface dominated HONO production; 2) soil HONO emission may become an important source in rural area with large areas of agricultural fields during fertilization period; 3) HONO greatly aggravated O$_3$ pollution in China. Our study has important implication on O$_3$ mitigation for policymakers. Controlling HONO production may provide an alternative pathway for O$_3$ mitigation. We have added the following discussion on the application from our study in the revised manuscript:

Line 492-504:

*"In addition, now most studies are focusing on VOCs and NO$_x$ reduction to achieve O$_3$ mitigation. However, O$_3$ formation showed non-linear relationship on VOCs and NO$_x$, making it difficult to decrease O$_3$ by solely reducing VOCs or NO$_x$. For instance, during COVID-19 lockdown period, O$_3$ showed an evident increase while NO$_x$ and VOCs*

*showed a decrease trend, highlight the complexity of $O_3$ mitigation (Zhao et al., 2020; Wang et al., 2022). Our results suggested HONO contributed significantly to $O_3$ production in China, and thereby, reducing HONO production may be an alternative way for $O_3$ control. As $NO_2$ heterogeneous reactions on the ground surface were important sources for HONO production, reducing $NO_x$ emissions would be beneficial for reducing HONO emissions. However, $NO_x$ reduction may also lead to more $O_3$ production if $O_3$ formation is in a VOC-limited regime, and hence the overall effects of $NO_x$ reduction on $O_3$ should be evaluated by chemical transport models. Moreover, soil HONO emissions may become an important source in rural area with large areas of agricultural fields. Decreasing soil HONO emissions is beneficial for $O_3$ pollution control, especially during fertilization period in June when the $O_3$ pollution was severe. Therefore, more environmental-friendly fertilization amount and fertilization mode should be investigated to decrease soil HONO emissions."*

L43: please spell out "SOA"

As suggested, "SOA" has changed to "secondary organic aerosol (SOA)".

L166-167: any reference for "a lifetime of 8 hours"?

For reference, we have added a study by Ma et al. (2020). We assumed a lifetime of 8 hours with the aim of considering the loss by transportation and deposition. If this loss was incorporated, the modeled PAN agreed very well with observed PAN in this campaign (Figure S1), suggesting this incorporation was reasonable.

[Figure]

*Figure S2. Averaged diurnal pattern of observed and modeled PAN if A first-order dilution loss term with a lifetime of 8 hours was incorporated.*

L251: this argument should be further justified.

As suggested, we have revised the corresponding texts:
Line 249-251:
*"In addition, considering that the observation period covered the fertilization period in June, the high HONO/$NO_x$ in our study may be partially explained by direct soil HONO*

*emissions around the sampling sites."*

In Fig.1, The website of TROPOMI NO2 data product should be provided.

We have provided the following website of TROPOMI NO$_2$ data product:
https://s5phub.copernicus.eu/dhus

Reference:
Li, L., Duan, Z. Y., Li, H., Zhu, C. Q., Henkelman, G., Francisco, J. S., and Zeng, X. C.: Formation of HONO from the NH3-promoted hydrolysis of NO2 dimers in the atmosphere, P Natl Acad Sci USA, 115, 7236-7241, 10.1073/pnas.1807719115, 2018.
Liu, J., Li, B., Deng, H., Yang, Y., Song, W., Wang, X., Luo, Y., Francisco, J. S., Li, L., and Gligorovski, S.: Resolving the Formation Mechanism of HONO via Ammonia-Promoted Photosensitized Conversion of Monomeric NO2 on Urban Glass Surfaces, Journal of the American Chemical Society, 145, 11488-11493, 10.1021/jacs.3c02067, 2023.
Ma, X. F., Tan, Z. F., Lu, K. D., Yang, X. P., Chen, X. R., Wang, H. C., Chen, S. Y., Fang, X., Li, S. L., Li, X., Liu, J. W., Liu, Y., Lou, S. R., Qiu, W. Y., Wang, H. L., Zeng, L. M., and Zhang, Y. H.: OH and HO2 radical chemistry at a suburban site during the EXPLORE-YRD campaign in 2018, Atmos Chem Phys, 22, 7005-7028, 10.5194/acp-22-7005-2022, 2022.
Wang, H., Huang, C., Tao, W., Gao, Y., Wang, S., Jing, S., Wang, W., Yan, R., Wang, Q., An, J., Tian, J., Hu, Q., Lou, S., Pöschl, U., Cheng, Y., and Su, H.: Seasonality and reduced nitric oxide titration dominated ozone increase during COVID-19 lockdown in eastern China, npj Climate and Atmospheric Science, 5, 24, 10.1038/s41612-022-00249-3, 2022.
Xia, D. M., Zhang, X. R., Chen, J. W., Tong, S. R., Xie, H. B., Wang, Z. Y., Xu, T., Ge, M. F., and Allen, D. T.: Heterogeneous Formation of HONO Catalyzed by CO2, Environ Sci Technol, 55, 12215-12222, 10.1021/acs.est.1c02706, 2021.
Xu, W., Kuang, Y., Zhao, C., Tao, J., Zhao, G., Bian, Y., Yang, W., Yu, Y., Shen, C., Liang, L., Zhang, G., Lin, W., and Xu, X.: NH3-promoted hydrolysis of NO2 induces explosive growth in HONO, Atmos. Chem. Phys., 19, 10557-10570, 10.5194/acp-19-10557-2019, 2019.
Zhao, Y., Zhang, K., Xu, X., Shen, H., Zhu, X., Zhang, Y., Hu, Y., and Shen, G.: Substantial Changes in Nitrogen Dioxide and Ozone after Excluding Meteorological Impacts during the COVID-19 Outbreak in Mainland China, Environmental Science & Technology Letters, 7, 402-408, 10.1021/acs.estlett.0c00304, 2020.